# Piezo1 Is Required for Myoblast Migration and Involves Polarized Clustering in Association with Cholesterol and GM1 Ganglioside

**DOI:** 10.3390/cells12242784

**Published:** 2023-12-07

**Authors:** Juliette Vanderroost, Thibaud Parpaite, Noémie Avalosse, Patrick Henriet, Christophe E. Pierreux, Joseph H. Lorent, Philippe Gailly, Donatienne Tyteca

**Affiliations:** 1de Duve Institute, UCLouvain, 1200 Brussels, Belgium; juliette.vanderroost@uclouvain.be (J.V.); noemie.avalosse@uclouvain.be (N.A.); patrick.henriet@uclouvain.be (P.H.); christophe.pierreux@uclouvain.be (C.E.P.); 2Institute of Neuroscience, UCLouvain, 1200 Brussels, Belgium; thibaud.parpaite@uclouvain.be (T.P.); philippe.gailly@uclouvain.be (P.G.); 3Louvain Drug Research Institute, UCLouvain, 1200 Brussels, Belgium; joseph.lorent@uclouvain.be

**Keywords:** lipid domains, calcium live cell imaging, Fura-2 polarization, lipid lateral diffusion, Piezo1 KO cells, mechanosensitive channels, PMCA, chemokinesis, chemotaxis, topotaxis

## Abstract

A specific plasma membrane distribution of the mechanosensitive ion channel Piezo1 is required for cell migration, but the mechanism remains elusive. Here, we addressed this question using WT and *Piezo1*-silenced C2C12 mouse myoblasts and WT and *Piezo1*-KO human kidney HEK293T cells. We showed that cell migration in a cell-free area and through a porous membrane decreased upon *Piezo1* silencing or deletion, but increased upon Piezo1 activation by Yoda1, whereas migration towards a chemoattractant gradient was reduced by Yoda1. Piezo1 organized into clusters, which were preferentially enriched at the front. This polarization was stimulated by Yoda1, accompanied by Ca^2+^ polarization, and abrogated by partial cholesterol depletion. Piezo1 clusters partially colocalized with cholesterol- and GM1 ganglioside-enriched domains, the proportion of which was increased by Yoda1. Mechanistically, Piezo1 activation induced a differential mobile fraction of GM1 associated with domains and the bulk membrane. Conversely, cholesterol depletion abrogated the differential mobile fraction of Piezo1 associated with clusters and the bulk membrane. In conclusion, we revealed, for the first time, the differential implication of Piezo1 depending on the migration mode and the interplay between GM1/cholesterol-enriched domains at the front during migration in a cell-free area. These domains could provide the optimal biophysical properties for Piezo1 activity and/or spatial dissociation from the PMCA calcium efflux pump.

## 1. Introduction

Piezo1 is a mechanosensitive ion channel expressed in a wide variety of cell types such as endothelial cells, innate immune cells and muscle cells, etc. It plays an essential role in the transduction of mechanical cues into biological responses, like cell cycle progression, cell proliferation, cell migration, and apoptosis, through the activation of intracellular signaling pathways. Mutations of the *Piezo1* gene have been linked to several diseases, such as generalized lymphatic dysplasia, hereditary xerocytosis, and diabetes [1]. The upregulation of *Piezo1* can also favor the onset and progression of cancer [2].

Calcium (Ca^2+^) is known to be one of the most important second messengers to spatially and temporally coordinate cell migration [3]. The role of Piezo1-mediated Ca^2+^ influx in cell migration has been demonstrated, although the reports are somewhat contradictory. In microglia, *Piezo1* deficiency reversibly enhances migration towards a chemoattractant [4], while in the absence of a chemoattractant, Piezo1 activation using the allosteric agonist Yoda1 [5] was reported to promote microglial chemokinesis [6]. In contrast, in keratinocytes, Piezo1 activation with Yoda1 induces cell retraction, which limits migration and wound closure [7]. In human umbilical vein endothelial cells, *Piezo1* knockdown prevents cell migration towards VEGF (vascular endothelial growth factor) [8]. On the other hand, in a highly metastatic breast cancer cell line, *Piezo1* knockdown was shown to promote migration in an unconfined environment, while impeding confined migration [9]. 

One could explain these apparent discrepancies by the intrinsic complexity of cell migration. Several distinctions, each with their own specific features, are made between the type of migration (mesenchymal vs. amoeboid and collective vs. single-cell migration), the way in which migration is induced by a specific signal (chemical cues, substrate and topographic changes), and the manner in which the signal is detected by the cell (membrane receptors, membrane deformations, mechanosensitive ion channels) [10]. Moreover, multiple migration-inducing cues can occur simultaneously and either converge or counteract each other, increasing the complexity of this process. Thus, based on the physiological and cellular context, Piezo1 could have different impacts on cell migration. 

Mechanosensitive ion channels form a family of ion gating channels that respond to direct mechanical force and convert it to a biochemical cascade through a process called mechanotransduction. Two non-mutually exclusive hypotheses for Piezo1 gating are currently considered: the “force-from-lipids” and the “force-from-filaments” models, requiring the contribution of the plasma membrane (PM) vs. the cytoskeleton or extracellular matrix (ECM), respectively [11]. The elucidation of the structure of Piezo1 paved the way for the emergence of the “force-from-lipids” model [12]. Piezo1 is a homotrimer complex with a propeller-like structure exhibiting a central ion-conducting pore surrounded by three large peripheral domains called blades. Each domain is comprised of a succession of 36 transmembrane helical units and is connected to long intracellular beams. The arc-shaped blades were shown to induce a local curvature of the lipid bilayer and are thought to act as the channel mechanosensory domains [13]. A study proposed that the peripheral blade–beam complex could transfer mechanical forces from the lipid membrane to the pore through a lever-like transduction pathway leading to the channel opening [14]. A more recent study directly visualized Piezo1 by labeling each subunit of the protein and using two super-resolution nanoscopic imaging methods (i.e., PALM and MINFLUX). With these approaches, the authors showed that the blades are significantly expanded at rest due to the bending stress exerted by the PM, and that the flexibility of these blades has important implications for the properties of force transmission from the membrane to the pore domain [15]. Piezo1 was shown to flatten upon activation, which can be mediated by membrane lateral tension [16,17]. Regarding the “force-from-filaments” model, it has been suggested that forces are transmitted directly through the binding of ECM/cytoskeletal elements [18], or indirectly through modifications of the bilayer tension via linkages between the PM and the cytoskeleton [19]. The most accepted theory is that the two models complement each other, and that lipids act synergically with cytoskeletal and ECM components to finely regulate Piezo1 in complicated physiological processes, such as cell migration. However, in bleb-attached patches devoid of a cytoskeleton, it has been shown that Piezo1 activation could still occur [19], suggesting a more predominant role of the lipid bilayer. Membrane lipids could either directly interact with Piezo1 in “lipid binding pockets” and/or provide appropriate biophysical properties (i.e., membrane tension and curvature), either globally or locally, via the formation of PM lipid domains [20]. In favor of the latter possibility, it has recently been shown that Piezo1 clusterizes and that the disruption of cholesterol (chol)-enriched domains reduces channel sensitivity and slows down its activation [21], suggesting an association between these domains and Piezo1. 

In the present study, we sought to better understand the implications of Piezo1 under different conditions of cell migration and its interplay with lipid domains. To address this question, we used mainly the murine C2C12 myoblast cell line since (i) it displays a high migratory capacity related to the myogenesis process [22]; (ii) it is known to migrate in a single-cell mode in vitro [22,23,24], making it more convenient to study, as migrating cells are clearly delineated; (iii) it requires Piezo1 for differentiation and fusion [25,26]; and (iv) it presents several types of lipid domains, characterized both in the resting state and upon migration [27]. Thus, chol and sphingomyelin (SM) accumulated at the migration front, while ganglioside GM1 was enriched at the rear of the cell. Moreover, those lipids were enriched in stable submicrometric domains, with a majority of SM/chol- and chol-enriched domains concentrating at the migration front opposite to the GM1-enriched domains, which are more abundant at the rear (Appendix A) [27]. Here, using these C2C12 myoblasts, we first determined the impact of Piezo1 activation with Yoda1 or silencing with specific small interfering RNA (siRNA) on migration using different assays with various migration-inducing cues (migration in Ibidi chambers and in transwell chambers, with or without a chemoattractant gradient). We also took advantage of a human epithelial kidney (HEK293T) cell line knocked out (KO) for *Piezo1*. In addition, these cells use a collective mode of migration, contrasting with the single-cell mode of migration of myoblasts. We further tested the cellular Ca^2+^ distribution in migrating myoblasts upon Piezo1 modulation. The interplay between membrane lipids and Piezo1 was then studied by evaluating (i) the effect of lipid-specific pharmacological agents on Piezo1 membrane distribution, polarization, and lateral mobility; (ii) the extent of spatial association between Piezo1 and SM-, chol-, or GM1-enriched domains; and (iii) the impact of Piezo1 activation, silencing, or deletion on lipid polarization, lateral mobility, and domain formation. We finally compared the distribution of Piezo1 to the plasma membrane Ca^2+^ ATPase (PMCA) efflux pump upon myoblast migration. Our data revealed, for the first time, the implication of Piezo1 in myoblast migration in a migration mode-dependent manner. Hence, the data showed the contribution of GM1/chol-enriched domains, which could provide optimal biophysical properties for Piezo1 activity and/or spatial dissociation from the PMCA Ca^2+^ efflux pump, thereby controlling Ca^2+^ influx and efflux.

## 2. Materials and Methods

### 2.1. Cell Culture and Chemical Treatments

C2C12 mouse myoblasts, HEK293T, and *Piezo1*-KO HEK293T cells (HEK P1KO) [28] were grown at 37 °C with 5% CO_2_ in Dulbecco’s Modified Eagle Medium (DMEM, Gibco^TM^, Waltham, MA, USA), containing 4.5 g/L D-glucose and 25 mM HEPES and supplemented with 10% Fetal Bovine Serum (FBS), penicillin (100 U/mL), and streptomycin (100 µg/mL). To activate Piezo1, the cells were incubated in a serum-free medium containing 0.25, 0.5, or 1 µM Yoda1 (Tocris Bioscience^TM^, Bristol, UK) for 30 min at 37 °C except otherwise stated. To deplete and subsequently replete membrane chol, the cells were incubated with 5 mM methyl-β-cyclodextrin (mβCD, Sigma-Aldrich, St. Louis, MO, USA) for 30 min, then with 2.5 mM chol loaded in mβCD (Sigma-Aldrich) for 1 h, as previously described [27]. To simultaneously activate Piezo1 and deplete membrane chol, the cells were incubated in a serum-free medium containing 0.5 µM Yoda1 and 5 mM mβCD for 30 min at 37 °C. To inhibit sphingolipid synthesis, the cells were incubated with 30 µM Fumonisin B1 (FB1, Sigma-Aldrich) for 48 h, as previously described [27]. 

### 2.2. Transfection of Piezo1 siRNAs or Piezo1 Fluorescent Constructs

To target the mouse *Piezo1* gene, two different siRNAs were used (siPiezo1 #1 and siPiezo1 #2, Appendix A), as well as a *Silencer™* Negative Control No. 1 siRNA (ThermoFisher, Waltham, MA, USA). The myoblasts were seeded in 6-well plates at a density of 10,000 cells/cm², then transfected overnight at 37 °C with 100 nM siRNA using the jetPRIME^®^ transfection reagent (Polyplus Transfection, Illkirch-Graffenstaden, France), following the manufacturer’s instructions. The cells were analyzed 72 h after transfection. To visualize Piezo1 in living cells, two different plasmids encoding a fluorescent Piezo1 were used (Piezo1-1591-GFP [19] and Piezo1-EGFP, with EGFP fused at the C-terminal of the protein). The myoblasts were seeded at a density of 10,000 cells/cm² in 24-well plates on uncoated or 10 µg/mL fibronectin-coated (Bovine Plasma Fibronectin, Gibco^TM^, Waltham, MA, USA) glass coverslips for the experiments on resting cells, or in Culture-Insert 2 Well dishes (Ibidi, Gräfelfing, Germany), for migration experiments. The cells were transfected overnight at 37 °C with 2 µg DNA using the jetPRIME^®^ transfection reagent at a DNA/jetPRIME^®^ ratio of 1:2, following the manufacturer’s instructions. The cells were analyzed 24 h post-transfection, and the transfection efficiency was visually estimated to be ~20%, which is consistent with previous reports of myoblast transfection with non-viral approaches [29].

### 2.3. Ibidi Chamber or Transwell Migration Assays

For migration in Ibidi chambers, myoblasts or HEK cells were seeded in Culture-Insert 2 Well dishes at a density of 20,000 cells/cm² or 70,000 cells/cm², respectively, then treated with Yoda1 for 30 min, then allowed to migrate for 5 h at 37 °C in serum-containing medium, as previously described [27]. For the transwell migration assay, the myoblasts or HEK cells were seeded in ThinCert^®^Tissue Culture Inserts with an 8 µm pore size (Greiner Bio-One, Kremsmunster, Austria) at the same density and allowed to migrate for 8 h at 37 °C, as previously described [27]. In the case of myoblasts, they were pretreated or not with Yoda1 before initiation of migration in the same conditions as in Ibidi chambers. The myoblasts migrated towards a gradient of 250 ng/mL insulin-like growth factor-1 (IGF-1, Sigma-Aldrich), while the HEK cells migrated towards a gradient of 10% FBS.

### 2.4. Calcium Live Cell Imaging

C2C12 or HEK cells were loaded with 2 µM Fura2-AM for 40 min in Krebs solution (133 mM NaCl, 3 mM KCl, 2.5 mM CaCl_2_, 1 mM MgCl_2_, 10 mM HEPES, 10 mM glucose). The Fura2-AM-loaded cells were placed on the stage of an upward epifluorescence microscope (Zeiss Axio Examiner, Oberkochen, Germany) and continuously superfused using Krebs solution. Fura2-AM was alternatively excited at 340 and 380 nm, and the ratios of the resulting images (340/380) were produced every second. The source of excitation was a xenon arc lamp, and the excitation wavelength was selected by a fast excitation switching filter (Lambda DG-4, Sutter Instrument, Novato, CA, USA). Emission was recorded at 510 nm, and the digital images were acquired through a 20× water immersion objective before being sampled at a 12-bit resolution by a fast-scan CMOS digital camera (Prime 95B, Teledyne Photometrics, Tucson, AZ, USA). All the images were background-subtracted and controlled by MetaFluor software (Molecular Devices, San Jose, CA, USA). Yoda1 (0.5 µM for C2C12 myoblasts or 20 µM for HEK cells) was applied for 90 s. A positive response was defined as a signal at least 6 times higher than the Δ_ratio_ standard deviation, determined from a 5 min recording of the baseline.

### 2.5. Fura2 Polarization

Within each set of experiments, a first set of slices was used as the internal control to validate the efficiency of *Piezo1* silencing using Ca^2+^ imaging, as described above. Slices not used for Ca^2+^ imaging were preincubated in a serum-free medium containing 0.5 µM Yoda1 for 30 min at 37 °C, then imaged to measure the Fura2 polarization. Individual migrating cells were imaged with a 40× water immersion objective. Using ImageJ software v1.54f, all the images were background-subtracted. Ratio images were thresholded, and the background was set to NaN (Not A Number value; in this way, the background pixels were not included in the mean grey value (MGV) measurement). An averaged plot profile line wide enough to include the whole cell was captured. The rear and front parts of each cell were defined as the first or last 18% of the overall length of each individual cell, respectively. For representation only (heatmap and mean profiles), each profile was linearly interpolated to match the mean cell length (using RStudio v2022.07.1 and “stats” package v4.3).

### 2.6. Focal Adhesion, Piezo1 and PMCA Immunofluorescence and Quantification

The myoblasts were seeded on glass coverslips for the experiments performed on resting cells, or in Culture-Insert 2 Well dishes for the migration experiments at a density of 20,000 cells/cm² treated with Yoda1, mβCD or FB1, and allowed to migrate for 5 h for the latter case. Cells were then fixed, permeabilized (or not), blocked, and immunolabeled with a monoclonal paxillin antibody (Merck Millipore, Burlington, MA, USA, dilution 1:200), a polyclonal Piezo1 antibody (Proteintech, Rosemont, IL, USA, dilution 1:50), and/or a monoclonal PMCA antibody (ThermoFisher, dilution 1:50) as previously described [27]. Focal adhesion (FA) number and distribution were quantified as described in Ref. [27]. The polarization of Piezo1 and PMCA was quantified using the ratio of the MGV at the migration front vs. the MGV at the rear, and Piezo1/PMCA colocalization was determined using green or red colocalization coefficients (CC), which are defined as the colocalized pixel count divided by the sum of the colocalized and the green or red channel pixel counts, with the advantage of taking into account the variability of the signal intensity between the two channels [27].

### 2.7. Live Cell Imaging of Plasma Membrane Cholesterol, GM1 Ganglioside, and Sphingomyelin, and the Quantification of Polarization and Distribution in Domains

The myoblasts were seeded on glass coverslips for the experiments performed on resting cells or in Culture-Insert 2 Well dishes for the migration experiments at a density of 20,000 cells/cm², while the HEK cells were seeded on fibronectin-coated glass coverslips at a density of 70,000 cells/cm². Myoblasts were treated or not with Yoda1, then the resting cells were immediately labeled, while the migrating cells were labeled after 5 h of migration. Chol labeling was performed with mCherry-Theta toxin fragment (Theta), GM1 with BODIPY-GM1 or Alexa Fluor 647-conjugated Cholera toxin B subunit (CTxB, Invitrogen, Waltham, MA, USA), and SM with BODIPY-SM or mCherry-Lysenin toxin fragment (Lysenin), using the same concentrations and timings as in Ref. [27]. The labeling was performed at 4 °C, and the cells were visualized with a Zeiss Cell Observer Spinning Disk (COSD) confocal microscope using a plan-Apochromat 63×/1.4 water-immersion objective, for the resting experiments, and with a Zeiss Laser Scanning Microscope LSM980 Airyscan 2 confocal microscope using a plan-Apochromat 63×/1.4 oil-immersion objective, for migration experiments. The polarization of the lipids was quantified using the ratio of the MGV at the front vs. the MGV at the rear, while the lipid domain proportion was quantified using fluorescence intensity profiles, as previously described [27]. To determine the number of lipid domains relative to the cell surface, the number of each type of lipid domain quantified in the “proportion” analysis was added and then divided by the cell surface, and the treated conditions were expressed as the percentage of the control. To determine lipid colocalization, red and green CC were used, as explained in Section 2.6.

### 2.8. Fluorescence Recovery after Photobleaching of Membrane Lipids and Piezo1

Fluorescence recovery after photobleaching (FRAP) was performed with the LSM980 Airyscan 2 confocal microscope using a plan-Apochromat 63×/1.4 oil-immersion objective. To study the lateral diffusion of the membrane lipids, the myoblasts were seeded in Culture-Insert 2 Well dishes at a density of 20,000 cells/cm², treated or not with Yoda1, migrated for 5 h at 37 °C, then labeled for chol (Theta) or for GM1 (CTxB) and photobleached at room temperature (RT). Four zones of 5 µm² were photobleached per migrating cell: one containing chol- or GM1-enriched domains and one representing the bulk membrane, at both the migration front and rear of the cell. A non-photobleached control zone was also defined to check for fluorescence stability. The photobleaching parameters were set so as not to exceed 60% of photobleaching to avoid cell toxicity, and these parameters were identical to those used in Ref. [27]. To study the lateral diffusion of Piezo1, the myoblasts were seeded in Culture-Insert 2 Well dishes at a density of 20,000 cells/cm², transfected overnight with a fluorescent Piezo1 construct (as explained in Section 2.2), treated or not with Yoda1, mβCD or the combination of both treatments, then photobleached at 37 °C. Two zones of 5 µm² were photobleached per myoblast: one containing Piezo1 clusters and one representing the bulk membrane. A non-photobleached control zone was also defined, and the photobleaching parameters were set as follows: 5 iterations at 80% laser intensity, with images taken every 8 s for 13 cycles after photobleaching. The mobile fraction and the half-time of fluorescence recovery (T1/2), which is defined as the time required for a bleached area to recover half of the maximal recovery, were then deduced using a one-phase association fitting [27].

### 2.9. Real-Time Quantitative PCR Analysis of Piezo1

Total RNA was extracted from myoblasts seeded in 6-well plates using 400 µL TRIzol reagent (ThermoFisher). Then, 200 µL chloroform was added, and the mixture was incubated for 15 min at RT. The lysates were centrifuged for 15 min at 4 °C and at 15,000× *g*. The upper phase was retrieved and precipitated with 500 µL isopropanol and 30 µg of GlycoBlue Coprecipitant (ThermoFisher) for 1 h at −80 °C. The solution was then centrifuged for 30 min at 4 °C at 15,000× *g* to pellet the RNA. The supernatant was eliminated and the pellet rinsed with 1 mL 70% cold ethanol, followed by a 10 min centrifugation at 4 °C at 15,000× *g*. The supernatant was eliminated and the pellet left to dry before resuspension in 11 µL of RNAse-free H_2_O_d_. The RNA concentration was evaluated using a NanoDrop8000 spectrophotometer (ThermoFisher). Reverse transcription was then conducted on 500 ng of total RNA using M-MLV Reverse Transcriptase (Invitrogen) and random hexamers, following the manufacturer’s instructions. Finally, real-time quantitative PCR was carried out on 15 ng cDNA using the KAPA SYBR FAST qPCR Master Mix 2× (Kapa Biosystems, Wilmington, MA, USA), following the manufacturer’s instructions, along with a Piezo1 primer pair (Appendix A). The data were analyzed using the ∆∆CT method, with *Gapdh* as the reference gene. The data are expressed as a base-2 logarithm of the fold change.

### 2.10. Data Presentation and Statistical Analyses

Data are represented as means of *n* independent experiments or means of x cells from *n* independent experiments ± SD. All statistical analyses were performed using GraphPad Prism 8.0.2. Statistical tests were performed when *n* ≥ 3. The normality of data was assessed using a QQ plot. For non-Gaussian unpaired data, a non-parametrical Kruskal–Wallis test, followed by Dunn’s multiple comparisons test, were performed to compare more than two groups. For (paired) Gaussian data, (paired) *t*-tests were used to compare two groups, and (paired) one-way ANOVA parametric test was performed to compare more than two groups. The latter was followed by Tukey’s multiple comparisons test to compare the mean of each column with the mean of every other column, or by Dunnett’s multiple comparisons test to compare the mean of each column with the mean of a control column. Finally, grouped samples were analyzed using a parametrical two-way ANOVA, followed by Sidak’s multiple comparisons test. Paired data are graphically represented by the same symbol shape, while the colors represent the treatment. Comparisons with the control value are indicated above the symbols, while comparisons between two or more groups are indicated with bars on top of the graphs. When *n* ≥ 3 and nothing is indicated on the graphs, the data are non-significant. *, *p*-value < 0.05; **, *p*-value < 0.01; ***, *p*-value < 0.001.

## 3. Results

### 3.1. *Piezo1* Silencing in Myoblasts and *Piezo1* Knockout in HEK Cells Decrease Migration in Ibidi and Transwell Chambers, While Piezo1 Activation in Myoblasts Acts in an Opposite Manner

To study the Piezo1 implication in cell migration, we used three migration assays, taking into account various migration-inducing cues [10]. The first assay takes place in Ibidi chambers, with a 2 well silicone insert that defines a cell-free gap. When the insert is removed, the cells can migrate in the cell-free area as diffusible cues, such as growth factors, are homogeneously diluted in the medium. The second assay is a transwell assay in which cells are seeded on the upper layer of a cell culture insert and then migrate through the confined 8 µm pores of a permeable membrane. In this case, the medium does not contain any chemoattractants and is the same in the upper and lower chamber. The third assay is also based on the transwell assay, but this time, the bottom chamber is filled with a chemoattractant, thereby defining a chemical gradient that stimulates the directed migration of cells through the permeable membrane.

Using these three assays, we found that Piezo1 activation with increasing concentrations of the specific allosteric activator Yoda1 did not significantly impact myoblast migration in the Ibidi chambers (Figure 1A), but increased myoblast migration, in a dose–response manner, in the transwell chambers (Figure 1B). Surprisingly, the addition of a chemoattractant gradient abrogated this effect and even decreased myoblast directed migration in transwell chambers upon Piezo1 activation (Figure 1C). *Piezo1* silencing using two validated siRNAs (siPiezo1 #1 and siPiezo1 #2, Appendix A) achieved the opposite results, with a decrease in myoblast migration in the Ibidi (Figure 1D) and transwell chambers (Figure 1E) and an increase in directed migration in the transwell chambers, when a chemoattractant gradient was formed (Figure 1F). Our results indicate that Piezo1 is involved in myoblast migration, but differentially based on the migration assay.

To further test this implication, we took advantage of an HEK293T cell line KO for *Piezo1* (HEK P1KO) [28]. Furthermore, HEK cells preferentially migrate in a collective mode of migration [30] in contrast to myoblasts, which migrate individually [27], allowing for the comparison of these two modes regarding the implication of Piezo1. These cells were, as expected, unresponsive to Yoda1 (Appendix A) and showed a strong reduction of migration in the Ibidi (Figure 1G) and transwell chambers (Figure 1H), similar to the *Piezo1* silencing noted in the myoblasts (Figure 1D,E). In contrast to the *Piezo1*-silenced myoblasts, HEK P1KO directed migration was decreased when a chemoattractant gradient was set within the transwell assay (Figure 1I), suggesting that Piezo1 implication in cell migration could also depend on the individual or collective mode of cell migration.

### 3.2. Piezo1 Activation Increases Calcium Levels, Particularly at the Front of Migrating Myoblasts

The above data indicated that Piezo1 was required for the migration of myoblasts and HEK293T cells in Ibidi chambers and transwells in the absence of a chemoattractant. We therefore explored the mechanism involved using Ibidi chambers, which are compatible with cell imaging. We started by visualizing Ca^2+^, an important second messenger that coordinates cell migration both spatially and temporally [3]. We used the membrane-permeant Fura2-AM, a ratiometric Ca^2+^ indicator whose excitation wavelength shifts from 380 to 340 nm when bound to cytosolic free Ca^2+^ [31], in the migrating C2C12 myoblasts (Appendix A). Upon *Piezo1* silencing with siRNAs, the number of C2C12 cells responding to Yoda1 was significantly decreased (Figure 2A). However, neither siRNA nor Yoda1 affected the mean baseline ratios in C2C12 myoblasts (Figure 2B). We then measured the 340/380 ratio on a rear-to-front axis of the migrating cells (Figure 2C) and transformed the data into a ratio at the front vs. the center of the cell to specifically determine the Ca^2+^ levels in different areas of the cell. In untreated C2C12 cells, a higher Ca^2+^ concentration was observed in the center part of the cell compared to the migration front, as reflected by the ratio <1 (Appendix A and Figure 2D,G, siCTL). Upon stimulation with Yoda1, the ratio significantly increased and was close to 1 due to an increase in Ca^2+^ levels at the cell front (Figure 2D,G, +Yoda1). To verify the implication of Piezo1 in this process, the same measurements were then performed on the *Piezo1*-silenced myoblasts. Under untreated conditions, higher Ca^2+^ levels in the center area compared to the front were still observed, and no changes occurred at the front upon Yoda1 stimulation (Figure 2E–G), confirming the implication of Piezo1 in this Ca^2+^ entry. Surprisingly, higher Ca^2+^ levels were observed upon Yoda1 at the rear, an observation which remains to be further analyzed. Altogether, this shows that Piezo1 activation can increase Ca^2+^ levels, particularly at the front in migrating myoblasts.

### 3.3. Piezo1 Gathers in Clusters and Polarizes at the Migration Front of Myoblasts, but Its Activation Does Not Modulate the Focal Adhesion Number or Distribution

We next visualized the Piezo1 organization and distribution at the cell surface using two complementary approaches: (i) immunolabeling with anti-Piezo1 antibody, which recognizes an extracellular epitope of Piezo1, thereby enabling the specific study of Piezo1 found at the PM; and (ii) live-cell imaging of two fluorescent Piezo1 fusion constructs (Piezo1-1591-GFP [19] and Piezo-EGFP), differing in the position of the reporter gene. Immunofluorescence experiments under non-permeabilized conditions revealed that the peripheral membrane pattern of Piezo1 labeling observed in the control cells had almost completely disappeared upon *Piezo1* silencing (Appendix A) or knockout (Appendix A), which was not the case under permeabilized conditions (Appendix A). Therefore, we prioritized experiments using this antibody on non-permeabilized cells. We then confirmed the presence of Piezo1 at the PM by fluorescent Piezo1 expression in resting myoblasts, while revealing their distribution in clusters, as shown on the XZ orthogonal reconstructions. Piezo1 was enriched around the cell protrusions, but also in the perinuclear region, as revealed on the basal sections (Figure 3A). The Piezo1 clusters were not affected by cell fixation (Appendix A), and they partially co-localized with the anti-Piezo1 signal, especially on permeabilized cells, but also partly on non-permeabilized cells (Appendix A), confirming that at least a portion of fluorescent Piezo1 clusters are distributed at the PM. When myoblasts were allowed to migrate for 5 h in Ibidi chambers, Piezo1 revealed by immunolabeling was polarized at the migration front, and activation by Yoda1 enhanced this event, as confirmed by the quantification of the ratio of Piezo1 fluorescence intensity at the front vs. the rear (Figure 3B,C). This enhanced polarization of Piezo1 seemed to result from a redistribution of Piezo1 from the rear to the front, as suggested by the absence of change in global fluorescence intensity between the control and Yoda1-stimulated cells (Appendix A).

Given that previous reports indicated a possible relationship between the FAs and Piezo1 [32,33,34], we immunolabeled migrating myoblasts for paxillin, a protein localized in the FA complex, and quantified various FA parameters, under control conditions or after Yoda1 pretreatment (Figure 3D–G). The results indicated that neither the FA number nor the distribution at the front vs. the center vs. the rear were affected by Piezo1 activation, suggesting that the Piezo1 implication in myoblast migration does not rely on FA regulation. However, this analysis was conducted on uncoated Ibidi chambers, which could impact the conclusion. For this reason, and because other reports also highlighted the influence of the ECM on Piezo1 mechanosensitivity [35], we cultured myoblasts expressing fluorescent Piezo1 on fibronectin-coated surfaces and compared the organization of Piezo1 clusters with the uncoated conditions. The results revealed that Piezo1 clusters were present on both uncoated and fibronectin-coated surfaces (Appendix A).

### 3.4. Cholesterol Depletion Impairs Piezo1 Clustering, and Reversibly Abrogates Piezo1 Polarization upon Myoblast Migration

We then evaluated the potential interplay with PM lipids by using two different pharmacological agents, i.e., methyl-β-cyclodextrin (mβCD), which depletes membrane chol, and fumonisin B1 (FB1), which inhibits sphingolipid synthesis. These agents were tested and validated for their specificity and non-toxicity in our previous study [27]. We first determined if and how these agents impacted Piezo1 clusters in resting cells expressing a fluorescent Piezo1 construct using fluorescence intensity profiles. The analysis was performed around the protrusions, as Piezo1 clusters at the PM seemed to be particularly abundant in these areas, which are also very thin, maximizing the chance that these clusters represent PM and not intracellular Piezo1. Piezo1 activation by Yoda1 increased the number of fluorescent peaks, whereas treatment with mβCD yielded the opposite result. FB1 treatment, on the other hand, did not show any effect (Figure 4A and Appendix A). These results suggest the specific implication of chol for Piezo1 clustering in resting myoblasts.

We then extended this analysis on the myoblasts in migration by quantifying Piezo1 polarization. As expected from Figure 3B, Yoda1 increased Piezo1 polarization at the migration front. Treatment with mβCD almost completely abrogated Piezo1 polarization, which became equally distributed at both extremities of the cell. Piezo1 polarization remained impaired even when Yoda1 was added on top of the mβCD treatment. This effect was reversible, since Piezo1 polarization was regained upon exogenous chol repletion after mβCD treatment. In regards to FB1 treatment, it did not affect Piezo1 polarization (Figure 4B,C). Altogether, these results suggest that chol is involved in both Piezo1 clustering and localization at the front upon migration.

### 3.5. Piezo1 Clusters Partially Associate with Cholesterol- and GM1-Enriched Domains in Migrating Myoblasts

To further assess the interplay between PM lipids and Piezo1, we investigated the spatial association between the Piezo1 clusters and the submicrometric lipid domains, previously identified at the myoblast surface [27]. To do so, myoblasts expressing a fluorescent Piezo1 were labeled at 4 °C with probes specific to chol, GM1, or SM under control conditions or after Piezo1 activation with Yoda1. This low temperature was required to avoid endocytosis. Nevertheless, we have verified that lipid domains were present upon labeling at room and physiological temperature, including upon endocytosis impairment [27]. The myoblasts were then visualized using confocal live cell imaging, analyzing them by drawing fluorescence intensity profiles at the migration front and by quantifying the colocalization coefficients of the entire cell or of the cell front. Images and profiles revealed that Piezo1 clusters partially associated with the chol-enriched domains (Figure 5A,B and Appendix A) and more importantly with the GM1-enriched domains (Figure 5C,D and Appendix A), but almost not with SM-enriched domains (Figure 5E,F and Appendix A). The colocalization coefficients confirmed these observations, regardless of the coefficient used (red or green) and no matter whether the analysis was performed on the entire cell (Figure 5G and Appendix A) or specifically at the migration front (Figure 5H and Appendix A). These results supported the implication of chol for Piezo1 regulation by showing their partial association, but also highlighted the association of Piezo1 with the GM1-enriched domains.

### 3.6. The Mobile Fraction of GM1 Associated with the Domains Is Lower than in the Bulk Membrane at the Front of the Migrating Myoblasts Activated for Piezo1

We previously indicated that chol and SM both polarized at the migration front, while GM1 polarized at the rear, and that treatment with mβCD specifically abrogated GM1 and SM polarization, which was not the case with FB1 (Appendix A) [27], showing the importance of chol for sphingolipid polarization. Since the PM lipids appeared to impact Piezo1 organization and localization, we then evaluated whether the opposite could also be relevant. Therefore, migrating myoblasts were treated with Yoda1 to activate Piezo1, and the lipid polarization was measured. We showed that treatment with increasing concentrations of Yoda1 did not affect chol, GM1, or SM polarization (Figure 6A–C), suggesting that lipid polarization was not dependent on Piezo1 activation.

We also evaluated the potential effect of Piezo1 activation on chol and GM1 membrane lateral diffusion, since both lipids were shown to spatially associate with Piezo1. To do so, using fluorescence recovery after photobleaching (FRAP), we compared the lateral mobility of chol and GM1, respectively decorated with Theta toxin fragment and the Cholera toxin B subunit, in areas containing domains vs. the bulk membrane at both the front and rear of the migrating myoblasts. We then determined the mobile fraction and half-time of fluorescence recovery for each zone (Figure 6D,E). For chol, no difference in mobile fraction could be seen between the chol-enriched domains and the bulk membrane, either at the front or the rear of the cells, whether Piezo1 was activated or not (Figure 6F). The same observation could be made for the GM1 mobile fraction under control conditions. However, upon Piezo1 activation, a significant difference was observed between the GM1-enriched domains and the bulk membrane at the migration front only, suggesting that Piezo1 activation could limit the mobility of GM1-enriched clusters at the front of the cells (Figure 6H). Regarding the diffusion time of chol and GM1, no significant difference was observed between conditions (Figure 6G,I). Altogether, these data indicate that Piezo1 activation neither modulated lipid polarization nor membrane chol lateral mobility, but could restrict GM1 lateral mobility, specifically in clusters and at the front of the migrating cells.

### 3.7. GM1-Containing Domains Are More Abundant upon Piezo1 Activation in Myoblasts and Are Impaired upon *Piezo1* Silencing or Deletion in Myoblasts and HEK Cells

Since GM1- and chol-enriched domains were partially associated with Piezo1 at the cell front, but Yoda1 did not affect GM1 polarization, we then determined whether Piezo1 activation could modulate the proportion of lipid domains at the surface of resting myoblasts. Orthogonal sections of resting myoblasts single-labeled for GM1 and treated with increasing concentrations of Yoda1 showed that GM1-enriched domains were more abundant, brighter, and therefore, more discernable, than those in the control. The morphology of the cell was also changed by Piezo1 activation, as it appeared to become polarized like a migrating cell, with a thicker and a thinner pole (Figure 7A). 

To determine the proportion of the different lipid domains per cell, the myoblasts were simultaneously triple-labeled for SM, chol, and GM1, and fluorescence intensity profiles were drawn, as described in Ref. [27]. In agreement with our previous data, we showed that, on resting myoblasts under control conditions, ~50% and ~25% of total domains were SM/chol/GM1- and chol-enriched domains, respectively. When the cells were stimulated with Yoda1, the proportion of GM1/chol- and GM1-enriched domains was significantly increased, while the proportion of chol- and SM/chol-enriched domains decreased (Figure 7B). Looking at the domain abundance instead of at the proportion, we found the highest increases for the GM1-containing domains, i.e., the GM1-, GM1/chol-, and SM/GM1-enriched domains (Figure 7C). In summary, Piezo1 activation seemed to specifically modulate the GM1-containing domains.

If Piezo1 activation specifically increased the abundance of GM1-enriched domains, the opposite effect should be observed in the *Piezo1*-silenced myoblasts. Since the transfection of siRNAs is not stable, thus precluding visualizing with certitude cells that effectively present the decreased expression of *Piezo1*, the selection of cells to be analyzed was based on the morphological criteria described above. Indeed, the polarization of the myoblasts was clearly visible upon stimulation with Yoda1 (Figure 7A), which could suggest the opposite result for myoblasts transfected with specific siRNAs targeting *Piezo1*. Therefore, the focus was centered on the non-polarized cells, which were shown to present a decreased abundance and enrichment of the GM1-enriched domains upon *Piezo1* silencing compared to the results for the control myoblast (Figure 7D,E). This effect seemed specific towards GM1, as the abundance of the SM-enriched domains appeared to remain unaffected (Figure 7F). To confirm this specificity with a more robust model, we visualized lipid domains in the HEK cell line KO for *Piezo1*. In line with previous observations, the GM1-enriched domains were almost completely abrogated in the HEK P1KO cell line (Figure 7G). In addition, the remaining GM1-enriched domains were dissociated from the chol-enriched domains, contrasting with the high colocalization between chol and GM1 in the HEK293T cells (Figure 7H). Conversely, the SM-enriched domains were present and colocalized with chol in the presence or absence of Piezo1 (Figure 7I). Altogether, these data indicate that the association of GM1-enriched domains with chol was specifically impacted by *Piezo1* silencing or deletion.

### 3.8. The Differential Mobile Fraction of Piezo1 between the Clusters and the Bulk Membrane Is Abrogated by Cholesterol Depletion in Myoblasts

Mechanistically, we propose that the lipid domains could provide appropriate biophysical properties to control Piezo1 distribution at the migration front. To test this hypothesis, we measured the lateral diffusion of fluorescent Piezo1 associated with clusters or with the bulk membrane under control conditions and after treatment with Yoda1 and/or mβCD in the resting myoblasts (Figure 8). Under control conditions and upon Piezo1 activation, we observed a restriction of Piezo1 lateral diffusion in clusters, with a ~50% mobile fraction compared to ~75% in the bulk membrane (Figure 8C,D and Appendix A). Chol depletion with mβCD abrogated this higher restriction in the clusters, reaching the same mobile fraction as in the bulk membrane (Figure 8D and Appendix A). The combination of mβCD and Yoda1 stimulation slightly restored the difference between the clusters and the bulk membrane for Piezo1-1591-GFP (Figure 8D). Conversely, for Piezo1-EGFP, the difference of the mobile fraction between the clusters and the bulk remained non-significant upon combining the treatments (Appendix A), and no significant changes could be observed for the diffusion time of Piezo1 in the clusters and in the bulk membrane, regardless of the treatment and the Piezo1 fluorescent construct (Figure 8E and Appendix A). These data reinforced the existence of the Piezo1 clusters with different properties than those in the bulk membrane and highlighted the implication of chol in their restriction of mobility.

### 3.9. Cholesterol Depletion Abrogates Both Piezo1 and PMCA Polarization and Spatial Dissociation upon Myoblast Migration

Finally, we addressed the physiological importance of this lipid-dependent Piezo1 clustering and polarization by testing the possibility that it could favor the spatial dissociation between Piezo1 and PMCA, thereby separating Ca^2+^ influx and efflux at the migration front. Therefore, we immunolabeled non-permeabilized migrating myoblasts for Piezo1 and PMCA under control conditions or after lipid modulation using mβCD or FB1, and we first verified the polarization of both proteins. Piezo1 polarized at the migration front as expected, which was abrogated upon chol depletion but non-significantly impacted by sphingolipid depletion (Figure 9A,B). PMCA was also polarized at the migration front, as previously shown in Ref. [36], and pharmacological treatments affected PMCA polarization to the same extent as for Piezo1 polarization (Figure 9A,C). The analysis of Piezo1/PMCA colocalization coefficients and fluorescence intensity profiles showed that, despite their similar polarization, Piezo1 and PMCA were almost completely dissociated under control conditions (Figure 9D,E). However, after mβCD or FB1 treatment, the colocalization between the two proteins strongly increased (Figure 9D,F,G), suggesting the importance of PM lipids for protein dissociation.

## 4. Discussion

### 4.1. Piezo1 Differential Implication in Cell Migration Based on Migration Mode

The wide variety of migration-inducing cues has made it necessary to classify migration in different modes in order to better understand how cells initiate movement based on these signals [10,37]. Therefore, we took care to compare the implication of Piezo1 in myoblast migration through multiple assays that differ in migration-inducing cues and reproduce specific features of chemokinesis, topotaxis, and chemotaxis. Migration in the Ibidi chambers resembled chemokinesis, as cell translocation was stimulated by uniformly distributed chemical cues, while the transwell chambers simulated topotaxis, since migration was stimulated by the topographic changes represented by the porous membrane. Since a gradient of diffusible cues is responsible for chemotaxis, the addition of a chemoattractant gradient in a transwell assay allowed for studying the combination of topotaxis- and chemotaxis-mediated migration. Based on this, we showed here that Piezo1 activation with Yoda1 did not impact chemokinesis, specifically enhanced topotaxis, but surprisingly decreased myoblast migration when topotaxis and chemotaxis were simultaneously stimulated. In *Piezo1*-silenced myoblasts using siRNAs, chemokinesis and topotaxis were both strongly impaired, but migration was restored and even increased upon the double stimulation of topotaxis and chemotaxis. The same results were observed in *Piezo1* KO HEK cells, except in the latter case, where migration was still impaired. Our data highlighted that Piezo1 could differentially control cell migration, depending on the mode of migration and the cell type. A differential implication of Piezo1 has similarly been evidenced in breast cancer cells. However, in contrast to our results, the authors showed that *Piezo1* silencing enhances migration under chemokinesis conditions and decreases migration in a confined environment with a chemotactic gradient [9].

Although Yoda1 did not significantly impact myoblast migration in the Ibidi chambers, *Piezo1* silencing strongly impaired it, suggesting the implication of Piezo1 in chemokinesis. This might be surprising at first glance, since Piezo1 is a mechanosensitive channel, and chemokinesis does not rely on mechanical cues, but on the chemical signals. Indeed, in this mode of migration, diffusible chemical cues are sensed by the cell through their binding to receptors, which ultimately activates intracellular signaling cascades to transmit the signal, initiating cell motility. However, it has previously been reported that several mechanosensitive channels from the transient receptor potential (TRP) family play an essential role in chemical-induced migration [38]. Moreover, upon the execution of the signal, cells do undergo massive morphologic changes, i.e., protrusion of the leading edge, its adhesion to the substrate, the translocation of the cell body, and the retraction of the rear [39], during which many forces and deformations are applied to the PM. In collective cell migration, the generation of local physical forces in the cell sheet produces a global state of tensile stress [40]. These physical constraints could open stretch-activated channels, such as Piezo1, supporting a reinforcement role rather than the initiation of chemokinesis, which is consistent with the fact that Yoda1 did not affect chemokinesis in the myoblasts.

Importantly, our chemokinesis assay was performed on uncoated Ibidi dishes and therefore, restrained FA maturation [41]. This has two implications. First, it could explain why Piezo1 activation did not impact either FA number or distribution in the myoblasts, contrasting with previous reports indicating Piezo1 association with integrins in FAs and their implication in their maturation and subsequent disassembly [33,34]. In MDCK epithelial cells, Piezo1 activation with Yoda1 treatment enlarges FAs on stiffer substrates, but has no effect on the formation of FAs on soft substrates [42]. As ECM components are key regulators of substrate stiffness [43], this highlights the importance of appropriate surface-coating to study the interplay between Piezo1 and FAs. Second, these weaker FAs could lead to a switch of migration mode, from a mesenchymal single-cell mode, which relies on strong adhesion with the ECM through FA complexes [39], to amoeboid migration, which would rely less on FAs. High plasticity in the migration modes in response to changes in cell confinement and adhesion has been described in the literature, especially in the mesenchymal-amoeboid transition [44], which has been observed to occur in satellite cells [45]. 

In contrast to chemokinesis, it is not surprising that Piezo1 activation or silencing/deletion strongly affected topotaxis, a mode of migration in which cells sense the topographical features of the surrounding microenvironment, such as pores or tunnel-like confined trails. Cells adapt their shape to the available geometry, and this deformation is notably sensed by mechanically coupled elements located at the cell surface [10]. Accordingly, it has been suggested that a confinement-driven force was transmitted via the activation of Piezo1, thereby mobilizing the cell [46]. Moreover, we showed that Yoda1 increased myoblast topotaxis to a great extent, and one could wonder how Piezo1 could still be locally activated at the migration front if it was pre-activated. Molecular dynamics simulations of mouse Piezo1 helped to construct a model suggesting that Yoda1 does not change Piezo1 conformation towards the open state, but rather acts as a molecular wedge facilitating force-induced conformational changes, effectively lowering of its mechanical threshold for activation [5]. As topotaxis relies on a more direct stimulation of Piezo1 than chemokinesis, it would be expected that there would be a higher impact on migration in the context of topotaxis upon the silencing or deletion of *Piezo1*. We observed this for the HEK collective migration. However, in myoblasts, the decrease in migration is similar for both chemokinesis and topotaxis, which could be explained, at least partially, by a compensatory mechanism involving other mechanosensitive channels, such as TRPV2, TRPM7, and TRPC1, shown to be expressed in the myoblasts. TRPC1 has also been shown to regulate C2C12 myoblast migration by the activation of calpains [26,47,48]. In HEK cells, the implication of the TRPC1, TRPC3, and TRPC7 channels in the store-operated Ca^2+^ entry has been evidenced [49], and TRPV4 was shown to participate in directional persistence and FA dynamics at the rear of the migrating HEK cells, possibly by forming a functional complex with TRPC1 [50].

Finally, a combination of cell confinement (topotaxis) and an IGF-1 gradient (chemotaxis) impaired myoblast migration when Piezo1 was activated. When *Piezo1* was knocked down, leading the cell to partially lose its ability to sense topographic changes, IGF-1 stimulation increased myoblast migration. Somehow, these signals counteracted each other in detriment to myoblast migration, forcing the cell to integrate the conflicting signals and subsequently, to prioritize [10]. Piezo1-mediated Ca^2+^ entry was shown to activate the RhoA/ROCK pathway in C2C12 myoblasts, which ultimately phosphorylates the myosin light chain 2 and promotes actomyosin assembly and contractility [26]. In the case of IGF-1, it binds to its tyrosine kinase receptor IGF-1R and activates the phosphoinositide 3-kinase pathway, which recruits and activates the Rho GTPases Rac1 and Cdc42 that regulate actin polymerization at the migration front [51,52]. In the C2C12 myoblasts, IGF-1 was shown to enhance migration by increasing calpain activity, potentially via the translocation of TRPC1 from a pre-existing intracellular pool to the PM [47]. The hypothetical compensation of *Piezo1* silencing by TRPC1 activity, enhanced by its recruitment through IGF-1 stimulation, could explain our results, but would require more investigation. When it comes to the migration of the HEK cells in the case of this double stimulation, Piezo1 seems to be more essential, as the migration of HEK P1KO cells remained lower than that of the control, despite the chemoattractant stimulation. However, the migration appeared higher than did the topotaxis alone, suggesting that chemical stimulation partially rescued the deletion of *Piezo1*. This could suggest that Piezo1 is more involved in collective migration than in single-cell migration, but this possibility should be considered very carefully, as these are two different cell types, and *Piezo1* was only knocked down in the myoblasts, opposite to the KO in HEK cells. Reports of a collective migratory behavior for myoblasts in specific experimental conditions underly the future possibility of comparing the implication of Piezo1 in single-cell and collective migration using the same cellular model [53,54].

### 4.2. Piezo1 and Ca^2+^ Polarization at the Migration Front upon Activation

Our analysis of Ca^2+^ distribution using Fura2-AM in migrating myoblasts revealed that Piezo1 activation increased Ca^2+^ levels, particularly at the front of the cell. As we further showed that Piezo1 polarized at the migration front, this accumulation could partly occur from Ca^2+^ entry at the front, but we cannot exclude the contribution of intracellular stores, as Yoda1 was also reported to release Ca^2+^ from the endoplasmic reticulum [55]. According to our observations, Piezo1 was also shown to polarize at the cell front in collective-migrating keratinocytes [7]. Conversely, previous studies have reported Ca^2+^ accumulation at the rear of migrating eosinophils and fibroblasts [56,57], as well as Piezo1 distribution at the posterior end of migrating fibroblasts and single-migrating keratinocytes [7,34]. Nevertheless, Ca^2+^ signals or microdomains, called Ca^2+^ flickers, were shown at the leading edge of the migrating fibroblasts. These flickers were proposed to act as central players in the spatiotemporal orchestration of cell migration [3,57,58], and their activity was shown to be coupled with membrane tension via TRPM7 [57]. Furthermore, Piezo1 activation in response to internal cell-generated forces forms spatially restricted Ca^2+^ flickers at the migration front of the fibroblasts [59]. 

### 4.3. Piezo1 Association with Specific Lipid Domains at the Migration Front

Besides Ca^2+^ level increase and Piezo1 polarization at the cell front upon Piezo1 activation, we also revealed the polarization of chol and SM [27], but only the former showed partial association with Piezo1, suggesting a specific Piezo1–chol interaction. In support of this possibility, computational studies identified at least 58 chol-recognition motifs within the *Piezo1* sequence. Some mutations of these domains were shown to influence the channel function, but most mutants were WT-like, suggesting that the chol effects are not mediated only by direct interactions [60,61]. Another possibility suggests that the membrane chol could regulate Piezo1 polarization at the front by providing appropriate membrane tension and curvature. Accordingly, we showed that upon myoblast migration, chol depletion abrogated Piezo1 polarization at the front. Chol is known to be a key regulator of membrane tension and to polarize at the migration front, where a transient increase in membrane tension contributes to the initiation of the lamellipodial protrusion [27,62,63]. Increased membrane tension induced by the pharmacological modulation of membrane lipids has been shown to activate multiple types of mechanosensitive channels [64]. Another study showed that increased membrane tension due to lipid peroxidation activates Piezo1 and TRP channels [65]. The contribution of membrane curvature in the distribution of Piezo1 is supported by the observation in the HeLa cells that membrane protrusions at the migration front, which are highly curved structures [66], are enriched in activated Piezo1 [17]. Accordingly, in the red blood cells, Piezo1 resides mainly in high-curvature areas, where chol-enriched domains gather upon red blood cell stretching [67] and are increased in areas of low curvature in response to Yoda1 stimulation [68]. We did not study Piezo1 activity here, but it was shown that chol depletion also reduces channel sensitivity, causing an increased channel latency of the response, slowing down its inactivation [21].

Nonetheless, such a role of chol cannot support the formation of Ca^2+^ microdomains and Piezo1 clusters in a dynamic manner. Piezo1 dynamics was previously studied, as it was claimed that the flat conformation of activated Piezo1 favors lateral diffusion, which could lead to more clusters of sufficient size to be resolved by confocal microscopy [17,68], and that Piezo1 could display transient spatial restriction, as suggested by the Piezo1-mediated Ca^2+^ flicker spatial restriction at the migration front [59]. We showed here that the formation of Piezo1 clusters at the cell front required chol and GM1, as supported by the following lines of evidence. First, chol depletion disrupted the Piezo1 clusters and abrogated the differential mobile fraction of the Piezo1 clusters compared to the bulk membrane by increasing the lateral mobility of the former. Second, we revealed the preferential spatial association between the Piezo1 clusters and the GM1-enriched domains, but also, and to a lesser extent, its association with the chol-enriched domains, but not with the SM-enriched domains. The study of Martinac et al. on HEK cells corroborates our findings, as they similarly showed that chol levels in the PM regulate the degree of Piezo1 clustering by affecting the density of clusters, the spatial association between Piezo1 and GM1, and diffusion of Piezo1 in chol-depleted cells [21]. Third, Piezo1 activation by Yoda1 induced a differential mobile fraction of GM1, associated with domains or present in the bulk, specifically at the cell front. Fourth, among the different types of lipid domains present at the myoblast surface, two were preferentially increased upon treatment with Yoda1, including those mainly enriched in GM1 and preferentially associated with the rear and those coenriched in GM1/chol and present at the front. The reason behind the increase in the GM1-enriched domains at the rear upon Piezo1 activation remains to be elucidated. In contrast, *Piezo1* silencing or deletion decreased the abundance and enrichment of the GM1-enriched domains. Fifth, anincrease in the GM1/chol-enriched domains upon Piezo1-mediated Ca^2+^ influx was similarly observed in red blood cells upon stretching and Piezo1 chemical activation [69].

### 4.4. Lipid-Dependent Spatial Dissociation of Piezo1 and PMCA at the Front

We also showed that the efflux pump PMCA polarized at the migration front as Piezo1, but that the two proteins were dissociated. This dissociation was abrogated upon chol and sphingolipid depletion. Based on several indications from our group, including (i) Piezo1 association with GM1/chol-enriched domains and its dissociation with the SM-enriched domains shown here, (ii) the polarization of SM/chol-enriched domains at the myoblast migration front [27], and (iii) the increase in SM/chol-enriched domains upon PMCA-mediated Ca^2+^ efflux during red blood cell deformation [69], we hypothesize that SM/chol-enriched domains could specifically regulate PMCA. This hypothesis will need to be tested in the future. From a mechanistic point-of-view, SM/chol-enriched domains could regulate PMCA localization and/or activity by providing appropriate fluidity [70,71,72]. In support of this hypothesis, a front-to-rear gradient of PMCA distribution, with a higher activity at the front, was evidenced in HUVEC cells [36], and chol is known to be a strong regulator of membrane fluidity [73]. This could indirectly involve phosphatidylinositol 4,5-bisphosphate (PIP2) [74]. Indeed, Ca^2+^ was shown to induce the clustering of PIP2 in the lipid bilayers through the interaction with its negative charges [75], and PIP2 was shown to bind with and to be a major activator of PMCA [74]. As SM-enriched domains at the outer PM leaflet were shown to be required for PIP2 enrichment at the inner PM leaflet in HeLa cells [76], these domains could be simultaneously recruited at the PMCA location, thereby modulating membrane fluidity.

## 5. Conclusions

Our data underlined the fact that Piezo1 is implicated in the single migration of myoblasts and the collective migration of HEK cells. For the myoblasts, this involves a differential implication of Piezo1, depending on the mode of migration stimulated (chemokinesis, topotaxis, or the combined stimulation of topotaxis and chemotaxis). Moreover, in a model of myoblast chemokinesis, the contribution of Piezo1 involves its clustering and polarization at the migration front, in association with the GM1- and chol-enriched domains, which could provide the appropriate biophysical properties to spatially control Piezo1, but this remains to be demonstrated. This could contribute to myoblast migration by regulating Ca^2+^ fluxes through the spatial dissociation between Piezo1 and PMCA, which also remains to be tested. 

## Figures and Tables

**Figure 1 cells-12-02784-f001:**
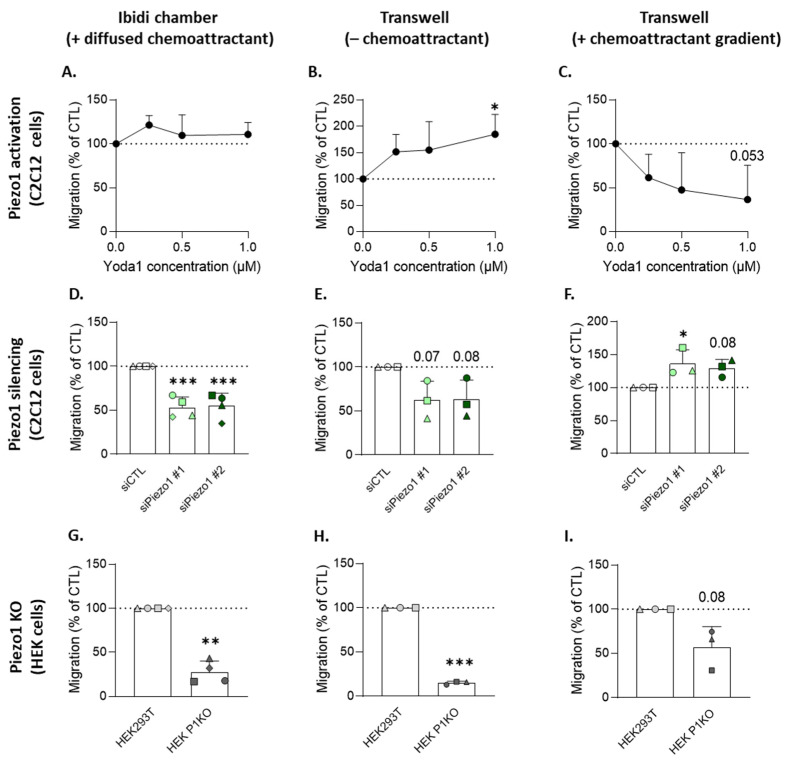
Piezo1 activation in the myoblasts stimulates migration in the transwell chambers only in the absence of a chemoattractant gradient, while *Piezo1* silencing or deletion in the myoblasts and HEK cells acts in an opposite manner. (**A**–**C**) C2C12 myoblasts were pretreated with increasing concentrations of Yoda1, then allowed to migrate for 5 h in Ibidi chambers (**A**), or for 8 h through transwell filters, without a chemoattractant (**B**), or towards a 250 ng/mL IGF-1 gradient (**C**). (**D**–**F**) C2C12 myoblasts were transfected with a negative control siRNA (siCTL, white) or with two different siRNAs targeting *Piezo1* (siPiezo1 #1 and #2, green), then allowed to migrate under the same conditions as those used in (**A**–**C**). (**G**–**I**) Control (HEK293T, light grey) or *Piezo1* knockout HEK cells (HEK P1KO, dark grey) were allowed to migrate for 5 h in Ibidi chambers (**G**), or for 8 h through transwell filters, without a chemoattractant (**H**), or towards a 10% FBS gradient (**I**). Data are expressed as means ± SD (*n* = 3–4 independent experiments) and the control value is indicated by the dotted lines. One-way ANOVA was performed, followed by Dunnett’s multiple comparisons test and a one sample *t*-test. Paired data are graphically represented by the same symbol shape while the colors represent the treatments. The statistics above the columns refer to the corresponding control (*, *p*-value < 0.05; **, *p*-value < 0.01; ***, *p*-value < 0.001).

**Figure 2 cells-12-02784-f002:**
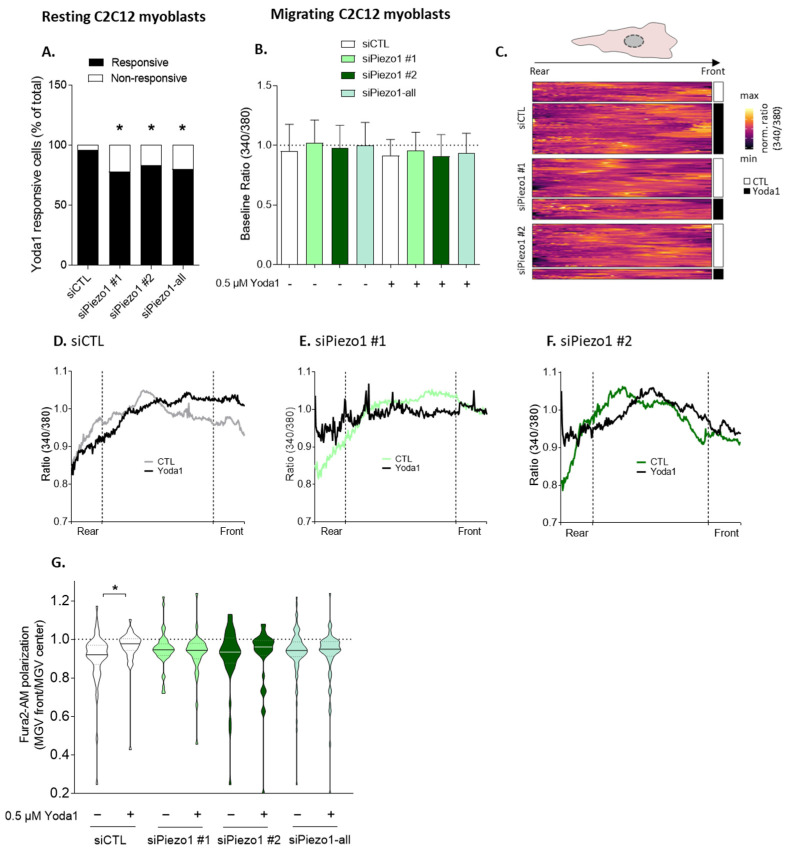
The increased Fura2 signal at the cell front upon Piezo1 activation by Yoda1 is abrogated by *Piezo1* silencing. Cells transfected with a negative control siRNA (siCTL, white) or with two different siRNAs targeting *Piezo1* (siPiezo1 #1 and #2, green) were left untreated or stimulated with 0.5 µM Yoda1 (black) for 90 s under resting conditions (**A**), or for 30 min and allowed to migrate for 5 h (**B**–**G**), then incubated with Fura2-AM and imaged at 340 and 380 nm with an epifluorescence microscope. The data obtained with siPiezo1 #1 and #2 were then pooled (siPiezo1-all). (**A**) The proportion of cells showing a Ca^2+^ response after 90 s of stimulation with 0.5 µM Yoda1. Significant decrease upon *Piezo1* silencing observed, according to the Fisher test. (**B**) Global Fura2-AM ratio (340/380) in whole migrating C2C12 cells (*n* = 4 independent experiments). (**C**) Heatmap of the ratio (340/380) along the main axis of each individual migrating cell. (**D**–**F**) Mean ratio (340/380) along the main cell axis from rear to front in siCTL- (**D**), siPiezo1 #1- (**E**), or siPiezo1 #2-transfected cells (**F**). (**G**) Violin plot of the ratio of the front to the center part of each cell (*n* = 250 cells from 4 independent experiments). Dotted line indicate the no polarization value; Straight lines in the plots indicate medians, while dotted lines in the plots indicate the interquartile range, Evaluated by Kruskal–Wallis test, followed by Dunn’s multiple comparisons test. The statistics above the columns refer to the corresponding control while the statistics between different groups are indicated with bars on top of graphs (*, *p*-value < 0.05).

**Figure 3 cells-12-02784-f003:**
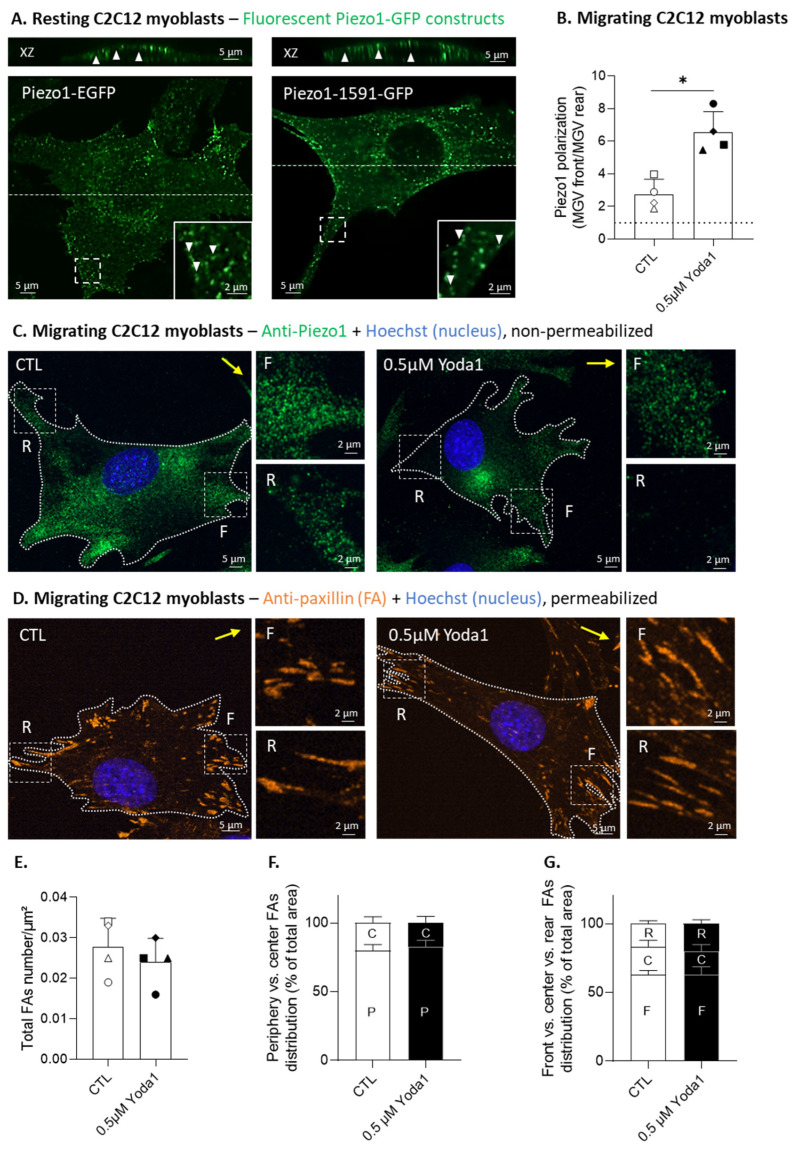
Piezo1 gathers in clusters, and Piezo1 activation favors its distribution at the cell front, but does not affect the number and distribution of focal adhesions. (**A**) Basal sections and orthogonal XZ reconstructions of C2C12 myoblasts transfected with two different fluorescent Piezo1 constructs and analyzed by vital confocal microscopy (Piezo1-EGFP and Piezo1-1591-GFP, representative of >5 independent experiments). Dotted lines—cell sections for the XZ reconstructions; arrowheads—Piezo1 clusters; insets—Piezo1 in cell protrusions. (**B**,**C**) Control (white) or cells pretreated with 0.5 µM Yoda1 for 30 min (black) were allowed to migrate for 5 h in Ibidi chambers, fixed and directly (immuno)labeled for Piezo1 (green) and a nucleus (blue), without cell permeabilization, to reveal the cell surface Piezo1, then they were visualized by confocal microscopy (**C**). Piezo1 polarization was quantified and expressed as the ratio of the mean gray value (MGV) at the front vs. the MGV at the rear ((**B**), *n* = 4 independent experiments). Dotted line—no polarization value. (**D**–**G**) Cells were treated and allowed to migrate, as in (**B**,**C**), then fixed, permeabilized, and (immuno)labeled for paxillin (orange) to visualize focal adhesions (FA) and for nuclei (blue) before being visualized by confocal microscopy (**D**). Quantification of total FAs number/µm² (**E**), FA center (C) or periphery (P) distribution (**F**), and FA front (F), center, or rear (R) distribution (**G**) was performed (*n* = 4 independent experiments). Distribution data are expressed as the % of total FAs area (means ± SD). Yellow arrows—direction of migration; F—migration front; C—center; R—rear. Evaluated by a Paired *t*-test and two-way ANOVA, followed by Sidak’s multiple comparisons test. Paired data are graphically represented by the same symbol shape while the colors represent the treatments. The statistics between different groups are indicated with bars on top of graphs (*, *p*-value < 0.05).

**Figure 4 cells-12-02784-f004:**
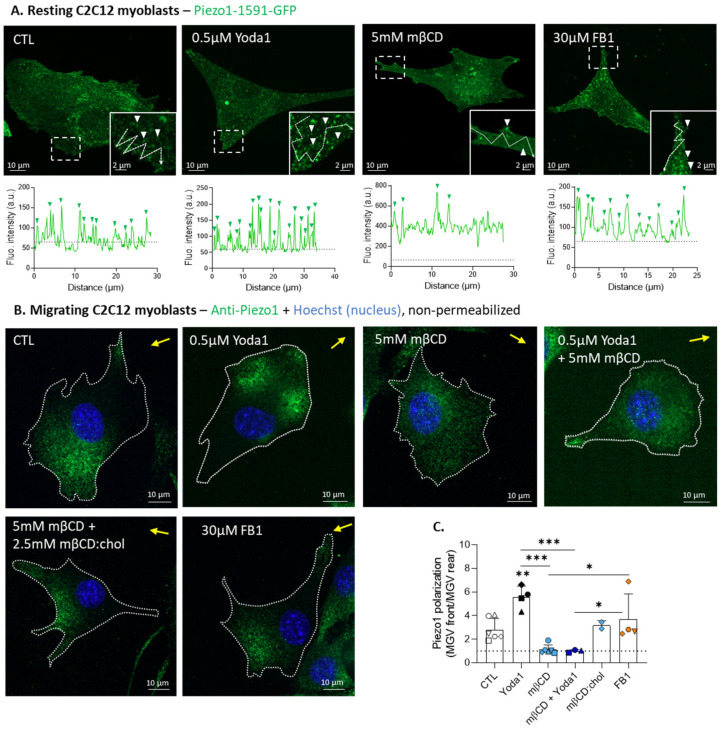
In contrast to sphingolipid depletion, cholesterol depletion impairs Piezo1 clustering and reversibly abrogates the increase in Piezo1 polarization by Yoda1. (**A**) Myoblasts were transfected with the fluorescent Piezo1-1591-GFP construct, then treated for 30 min with 0.5 µM Yoda1 or 5 mM mβCD, or for 48 h with 30 µM FB1, then visualized by vital confocal microscopy. Fluorescence intensity profiles were drawn along the cell periphery in protrusions (insets) to visualize Piezo1 cluster distribution. White arrowheads—Piezo1 clusters; dotted arrows—lines for profiles; dotted line on graphs, threshold; green arrowheads, fluorescent peaks. (**B**,**C**) Myoblasts were pretreated for 30 min with 0.5 µM Yoda1, 5 mM mβCD (followed or not by 2.5 mM chol repletion), a combination of both treatments, or for 48 h with 30 µM FB1. Then, the cells migrated for 5 h in Ibidi chambers, were fixed and directly (immuno)labeled for Piezo1 (green) and nuclei (blue), and visualized by confocal microscopy (**B**). Piezo1 polarization was quantified and expressed as the ratio of the MGV at the front vs. the MGV at the rear ((**C**), *n* = 4–6 independent experiments). Yellow arrow—direction of migration; dotted line on the graph—no polarization value. Evaluated using one-way ANOVA, followed by Tukey’s multiple comparisons test. Paired data are graphically represented by the same symbol shape while the colors represent the treatments. The statistics above the columns refer to the corresponding control while statistics between different groups are indicated with bars on top of graphs (*, *p*-value < 0.05; **, *p*-value < 0.01; ***, *p*-value < 0.001).

**Figure 5 cells-12-02784-f005:**
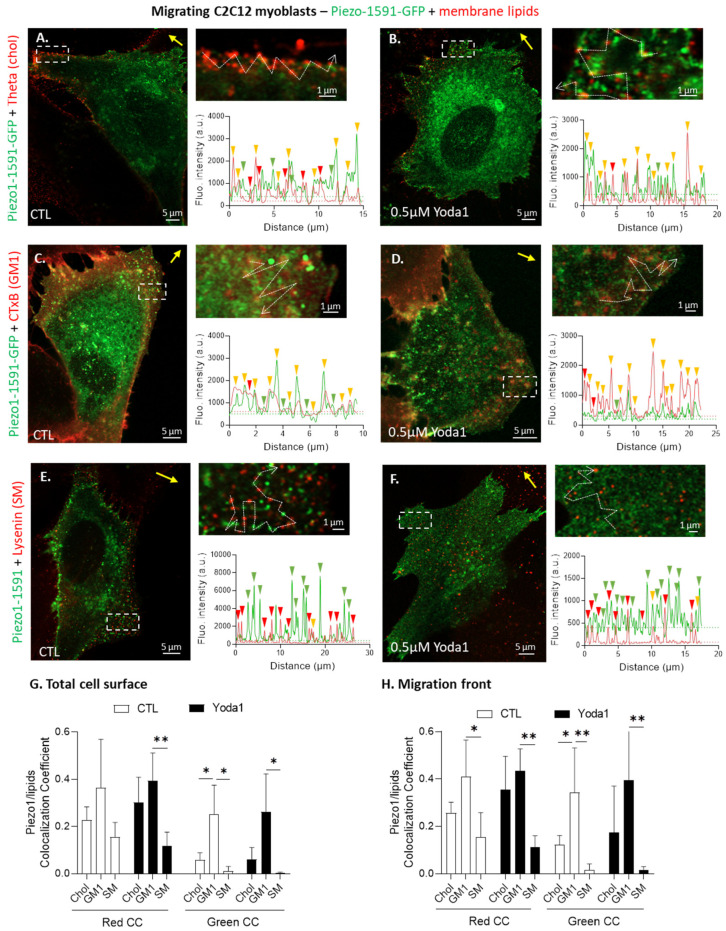
Piezo1 clusters mainly associate with GM1-enriched domains but also partly with the chol-enriched domains in migrating myoblasts upon activation with Yoda1 or not. (**A**–**F**) Cells were transfected with the fluorescent Piezo1-1591-GFP construct, left untreated (**A**,**C**,**E**) or pretreated with 0.5 µM Yoda1 for 30 min (**B**,**D**,**F**), then allowed to migrate for 5 h in the Ibidi chambers. The cells were next labeled for chol (Theta, (**A**,**B**)), GM1 (CTxB, (**C**,**D**)) or SM (Lysenin, (**E**,**F**)), visualized by vital confocal Airyscan microscopy in super-resolution mode, and fluorescence intensity profiles were drawn along the cell periphery to visualize the Piezo1 and lipid overlap. Yellow arrows—direction of migration; dotted arrows in insets—profiles; green or red arrowheads—Piezo1 or lipid enrichment; yellow arrowheads—colocalization; dotted lines on fluorescence intensity graphs—thresholds. (**G**,**H**) Quantification of the extent of colocalization between Piezo1 and membrane lipids at the total cell surface (**G**) or at the migration front specifically (**H**). Data are expressed as the sum of the colocalized pixel count divided by the sum of the colocalized and the red or green pixel counts, which correspond to the red or green colocalization coefficients (CC), respectively. Evaluated using two-way ANOVA, followed by Sidak’s multiple comparisons test. The statistics between different groups are indicated with bars on top of graphs (*, *p*-value < 0.05; **, *p*-value < 0.01).

**Figure 6 cells-12-02784-f006:**
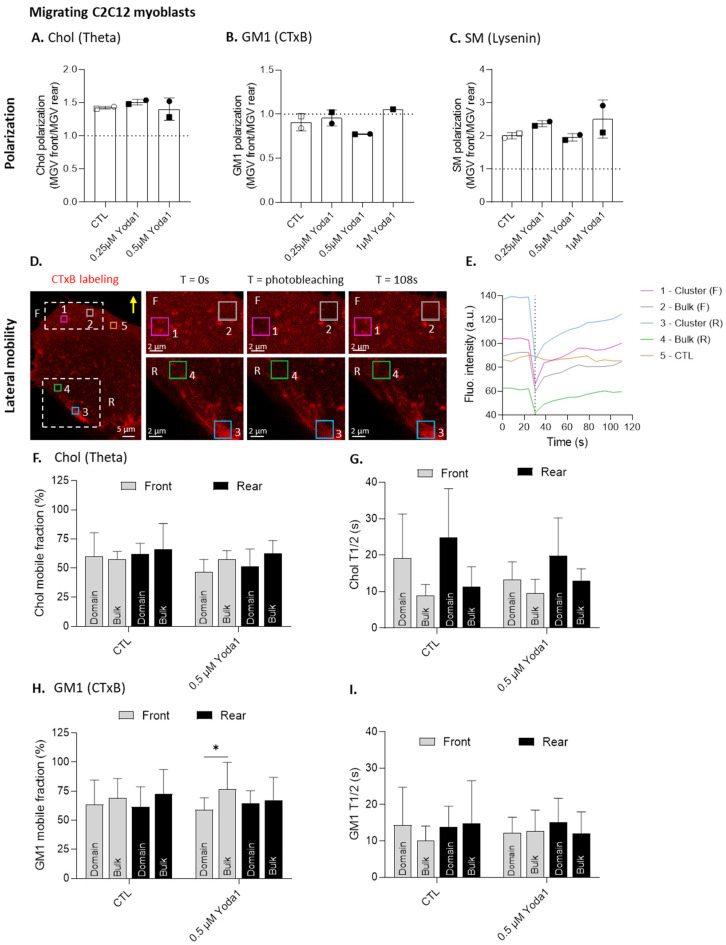
Piezo1 activation does not modulate either lipid polarization or membrane lateral diffusion, except for GM1 associated with domains that presents a lower mobile fraction, specifically at the front, when compared to the bulk membrane. Cells were left untreated (white), or pretreated with increasing concentrations ((**A**–**C**), black) or 0.5 µM (**F**–**I**) Yoda1 for 30 min, then allowed to migrate for 5 h in Ibidi chambers and single-labeled for chol (Theta), GM1 (CTxB) or SM (Lysenin) before being visualized by confocal microscopy (**A**–**C**) or analyzed by fluorescence recovery after photobleaching (FRAP) (**D**–**I**). (**A**–**C**) Quantification of chol (**A**), GM1 (**B**), or SM (**C**) polarization, expressed as the ratio of the MGV at the front vs. the MGV at the rear (*n* = 2 independent experiments). Dotted line—no polarization value. Paired data are graphically represented by the same symbol shape while the colors represent the treatments. (**D**–**E**) Representative FRAP experiment. Lipid lateral mobility was evaluated in four regions of interest represented by different colors: lipid domain or bulk membrane at the front and rear of the cell. A fifth region was not photobleached, but was used as the internal control (**D**). Yellow arrow—direction of migration. Data for fluorescence intensity over time were plotted on a graph (**E**). Vertical line—photobleaching time. (**F**–**I**) Quantification of chol (**F**,**G**) and GM1 (**H**,**I**) mobile fraction and half-time fluorescence recovery (*n* = 7–8 cells from two independent experiments for chol and 16–19 cells from four independent experiments for GM1). Data are expressed as means ± SD. Evaluated using two-way ANOVA, followed by Sidak’s multiple comparisons test. The statistics between different groups are indicated with bars on top of graphs (*, *p*-value < 0.05).

**Figure 7 cells-12-02784-f007:**
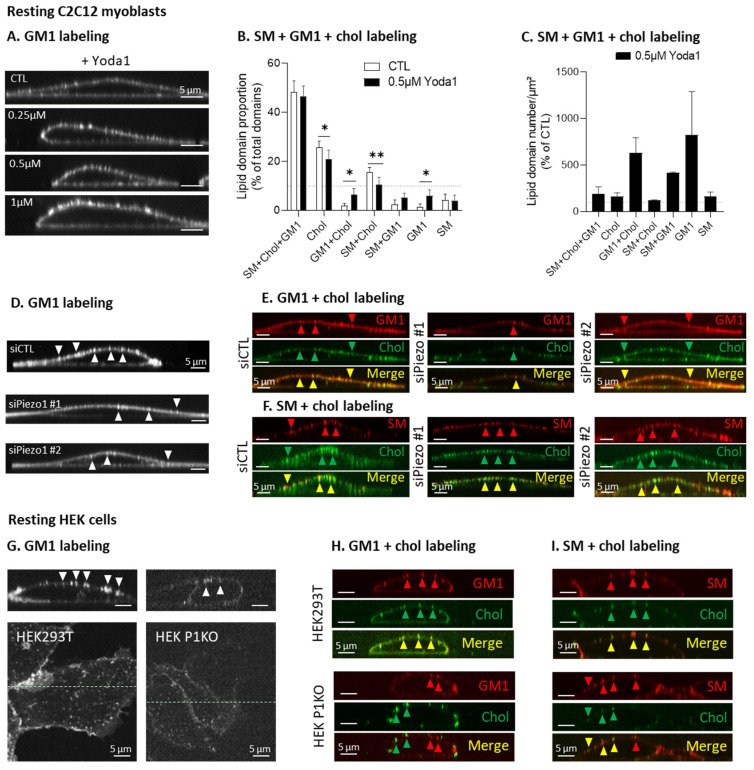
Piezo1 activation in myoblasts specifically increases the abundance of GM1-containing domains (i.e., GM1-, GM1/chol-, and SM/GM1-enriched domains), and *Piezo1* silencing in myoblasts or deletion in HEK cells acts in an opposite manner. (**A**) XZ orthogonal sections of resting C2C12 myoblasts single-labeled for GM1 (CTxB) under control conditions or after 30 min of pretreatment with increasing concentrations of Yoda1,and analyzed using vital confocal imaging. (**B**,**C**) Resting myoblasts were triple-labeled for chol, GM1, and SM (Theta, CTxB, and BODIPY-SM) under control conditions or after 30 min of pretreatment with 0.5 µM Yoda1. Then, lipid domain proportion (**B**) and lipid domain abundance at the cell surface (**C**) were quantified. Data are expressed as means ± SD (*n* = 6–7 cells from two independent experiments), evaluated by two-way ANOVA, followed by Sidak’s multiple comparisons test. The statistics between different groups are indicated with bars on top of graphs (*, *p*-value < 0.05; **, *p*-value < 0.01). (**D**–**F**) Orthogonal XZ reconstructions of myoblasts transfected with a control siRNA (siCTL) or with two different siRNAs targeting *Piezo1* (siPiezo1 #1 or #2), then single-labeled for GM1 (CTxB, (**D**)), double-labeled for GM1 and chol (CTxB + Theta, (**E**)), or double-labeled for SM and chol (BODIPY-SM + Theta, (**F**)) and analyzed by vital confocal imaging; representative of one experiment. (**G**–**I**) Basal sections and orthogonal XZ reconstructions of control (HEK293T) or *Piezo1* knockout HEK cells (HEK P1KO) seeded on fibronectin-coated coverslips and single-labeled for GM1 (CTxB, (**G**)), double-labeled for GM1 and chol (CTxB + Theta, (**H**)), or double-labeled for SM and chol (BODIPY-SM + Theta, (**I**)) and analyzed using vital confocal imaging; representative of two independent experiments. White arrowheads—GM1 clusters; red arrowheads—GM1 or SM clusters; green arrowheads—chol clusters; yellow arrowheads–colocalization.

**Figure 8 cells-12-02784-f008:**
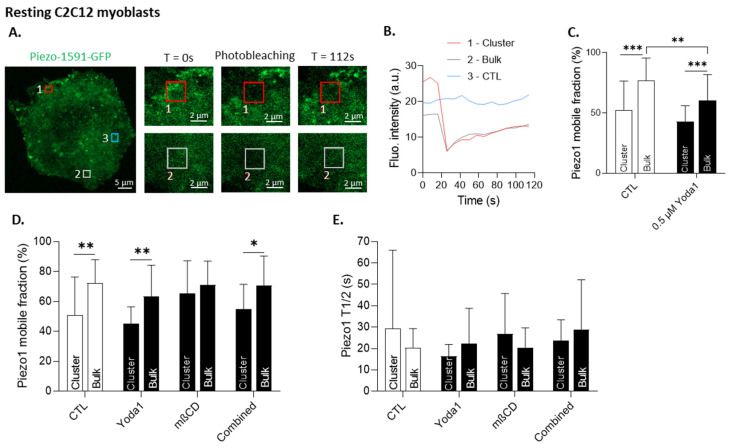
The membrane lateral diffusion of Piezo1 in the clusters is restricted compared to that of the bulk membrane, and this restriction is abrogated by chol depletion. Resting C2C12 myoblasts were transfected with the fluorescent Piezo1-1591-GFP construct, then treated for 30 min with 0.5 µM Yoda1, 5 mM mβCD, or a combination of both treatments and analyzed by FRAP. (**A**,**B**) Representative FRAP experiment. Piezo1 lateral mobility was evaluated in two regions of interest represented by different colors: Piezo1 clusters and Piezo1 in the bulk membrane. A third region was not photobleached, but was used as the internal control (**A**). Data for fluorescence intensity over time were plotted on a graph (**B**). (**C**–**E**) Piezo1 mobile fraction (**C**,**D**) (*n* = 33–34 cells from five independent experiments (**C**) and 12–18 cells from three independent experiments (**D**,**E**)) and half-time fluorescence recovery (**E**). Evaluated using two-way ANOVA, followed by Sidak’s multiple comparisons test. The control is indicated in white while black represents treated conditions. The statistics between different groups are indicated with bars on top of graphs (*, *p*-value < 0.05; ** *p*-value < 0.01, ***, *p*-value < 0.001).

**Figure 9 cells-12-02784-f009:**
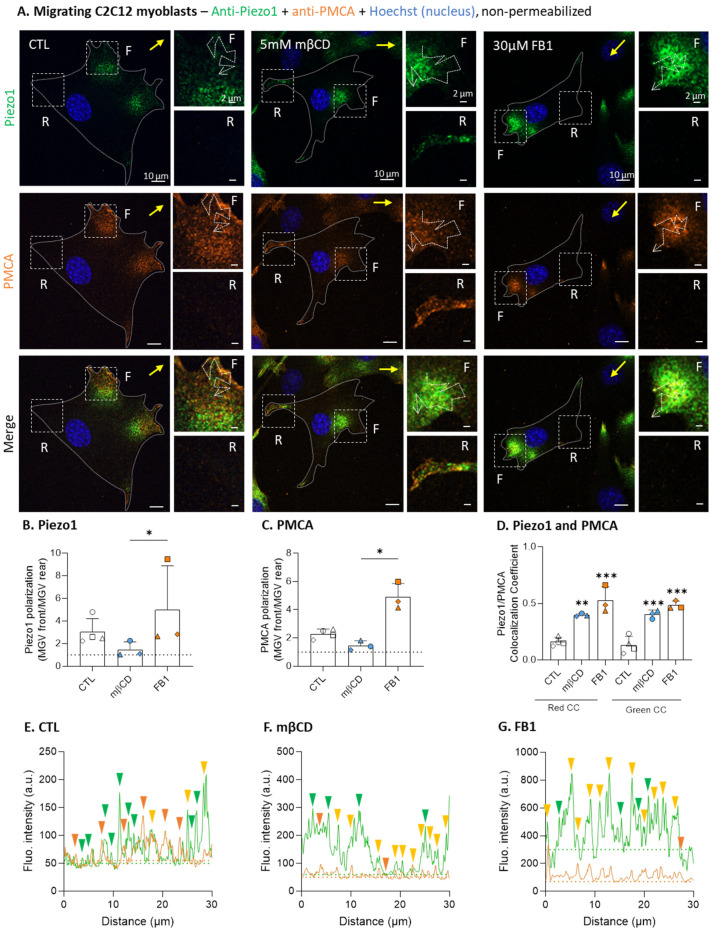
Piezo1 and PMCA are spatially dissociated at the cell front, which is impaired by cholesterol and sphingolipid depletion. (**A**) The cells were pretreated for 30 min with 5 mM mβCD or for 48 h with 30 µM FB1, then allowed to migrate for 5 h in Ibidi chambers, fixed and directly (immuno)labeled for Piezo1 (green), PMCA (orange), and the nucleus (blue) before being analyzed by confocal microscopy. Yellow arrows—direction of migration; F—migration front; R—rear. (**B**,**C**) Quantification of Piezo1 (**B**) and PMCA (**C**) polarization expressed as the ratio of the MGV at the front vs. the MGV at the rear (*n* = 3 independent experiments). (**D**) Quantification of the extent of colocalization between Piezo1 and PMCA at the total cell surface. The data are expressed as the sum of the colocalized pixel count divided by the sum of the colocalized and the red or green pixel counts, which corresponds to the red or green colocalization coefficient (CC), respectively (*n* = 3 independent experiments). Significance was evaluated using the Kruskal–Wallis test, followed by Dunn’s multiple comparisons test, and one-way ANOVA, followed by Tukey’s multiple comparisons test. (**E**–**G**) Fluorescence intensity profiles drawn at the migration front and represented by the dotted arrows in the insets at the front in (**A**) to visualize the Piezo1 and PMCA overlap. Green and orange arrowheads—Piezo1 and PMCA clusters; yellow arrowheads—colocalization; dotted lines—thresholds. Paired data are graphically represented by the same symbol shape while the colors represent the treatments. The statistics above the columns refer to the corresponding control while statistics between different groups are indicated with bars on top of graphs (*, *p*-value < 0.05; **, *p*-value < 0.01; ***, *p*-value < 0.001).

## Data Availability

Data are contained within the article and Appendix A.

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
