# Peer review of "Piezo1 Is Required for Myoblast Migration and Involves Polarized Clustering in Association with Cholesterol and GM1 Ganglioside"

_cells, 2023, doi:10.3390/cells12242784_

Round 1

Reviewer 1 Report

Comments and Suggestions for Authors

This manuscript is a very complete study that aims at the importance of Piezo1 in different mechanisms of cell motility and how it is related to some membrane lipids. The work is based on solid background very well presented in the introduction. Many varied techniques and methodologies were used and approach the problem from different perspectives. The photographs and graphics are excelent and illustrate the results adequately. Only one negative aspect is observed throughout the manuscript: in several passages it is not clear whether the results were obtained in the two cellular models, a myoblast line and a kidney line, that they use. To improve this aspect I suggest first mention in the abstract the cell lines that are used; then specify from the title of the subsection of the results section, whether the results described therein correspond to both cell lines or only to the myoblasts and finally, a final conclusion paragraph could be incorporated, this paragraph must includes the differences that were identified between both cell lines. Minor Consideration: Line 816-817 eliminate in the literature

Author Response

Brussels, November 27 2023

Dear Editor,

Let us first thank you very much for your feed-back on our manuscript cells-2639434, entitled ‘Piezo1 is required for myoblast migration and involves its polarized clustering in association with cholesterol and GM1 ganglioside’.

We are grateful for the analysis and the suggestions of the Reviewers. All comments were taken into serious consideration, as detailed hereafter in the point-to-point reply in attached file, and helped us to highly improve the manuscript.

We sincerely hope you will agree that all requests have been satisfactorily addressed.

Yours sincerely,

Prof. Donatienne Tyteca

DDUV Institute

UCLouvain, Brussels, Belgium

Reviewer 2 Report

Comments and Suggestions for Authors

In this study, Vanderroost et al. found that Piezo1 ion channel localization on the plasma membrane is crucial for cell migration. They also found that Piezo1 clusters, enriched in cholesterol and GM1 ganglioside, drive its polarization during migration, modulating calcium influx and efflux for effective cell movement. This sheds light on the role of specific membrane domains in regulating Piezo1 dynamics during cell migration. It is an interesting study, however, below are few of my suggestions:

1. Please discuss few recent studies in the discussion section.

2. Please add a note on novelty in both abstract and introduction sections.

3. I suggest to include a dedicated conclusion section.

4. Is it possible to include a Graphical Abstract?

5) Please cite few recent studies, especially introduction and discussion sections, as I could see most of the references are quite old.

6) Please proofread the manuscript to eliminate grammatical errors.  

Comments on the Quality of English Language

Slight editing of English language required

Author Response

Brussels, November 27 2023

Dear Editor,

Let us first thank you very much for your feed-back on our manuscript cells-2639434, entitled ‘Piezo1 is required for myoblast migration and involves its polarized clustering in association with cholesterol and GM1 ganglioside’.

We are grateful for the analysis and the suggestions of the Reviewers. All comments were taken into serious consideration, as detailed hereafter in the point-to-point reply in the attached file, and helped us to highly improve the manuscript.

We sincerely hope you will agree that all requests have been satisfactorily addressed.

Yours sincerely,

Prof. Donatienne Tyteca

DDUV Institute

UCLouvain, Brussels, Belgium

Reviewer 3 Report

Comments and Suggestions for Authors

Title: Piezo1 is required for myoblast migration and involves its polarized clustering in association with cholesterol and GM1 ganglioside. The authors demonstrate a Piezol 1 is a crucial factor for myoblast migration authors clearly asked relevant question to counterattack their hypothesis. So, based on this observation and their results persuade me to grant the permission for publication after major revision. I have little bit concern on this following point and expecting author should explain their point with experimental validation. 1) In Figure 3b Authors convincingly show that ratio of front to rare Piezo1 migration is higher in the yoda1 treated sample, however looking at figure 3c it is unclear whether the increase in the ratio is due to distribution of piezo1 from rare end to front end or is it due to depletion of piezo1 at rare end? 2) In text line 386-387 authors states “Results did not indicate significant changes in the presence of Piezo1 clusters (Figure S6)”. However, authors have not shown the quantitative data or statistics for images s6. 3) In figure 4a and s7 there is disparity in the section of the cell chosen for analysis. In certain images author selected front or rare or center without providing an explanation of how or why they selected particular section for analysis. 4) In Figure 7D-F, Author explained that Piezol 1 KO in HEK-293T cells completely abrogates the GM enriched domain as well co localization between GM1 and Cholesterol also hampered, which may be true however author need more validation in myoblast cells because their results in figure 1I does not match with actual finding. 5) Author need to explain more about their combine approach where, if combination of Yoda1 and mBCD treated at the same time and results are rescued when compared with mBCD treatment, means mBCD did not work or any other explanation about this? Can author repeat this experiment with endogenous piezol antibody. Overall, Author did some good experiment to prove their hypothesis however their clear vison from this article is not fulfilled with their experiment and need major revision to validate their hypothesis. This is minor comments: 1) in Figure 1a : Author used siRNA for Piezol 1, Author should include supplementary figure for SiRNA validation through immunoblot and also include the Nt SiRNA sequences in the file. 2) In all places author used short version of abbreviates like, PM, Chol, FM, ECM, it’s very frustrating for reader to understand their meaning, so my suggestion to author, put the abbreviation list in this article. 3) Why authors only used single cells of myoblast for their study. In conclusion, the authors convincingly demonstrate that Piezo1 is required for myoblast migration and involves its po- 2 larized clustering in association with cholesterol and GM1 ganglioside. My recommendation is that the authors should carefully review their data to ensure that their intended message. aligns with their actual finding.

Author Response

(The authors gave the same response as above.)

Round 2

Reviewer 3 Report

Comments and Suggestions for Authors

I appreciate the author, they worked carefully on the comments, and  i accepted the article in present form.